# Learning from Complaints: Adversarial Disentanglement for Robust Scalper Detection in E-Commerce Promotions

## Abstract

Identifying scalpers in e-commerce promotions is a critical challenge where *instance-dependent* label noise is pervasive: legitimate users with ambiguous patterns (e.g., *frequent on-the-hour purchases of high-subsidy items* and *orders shipped to non-habitual addresses*) are often misclassified as scalpers, leading to some user complaints and operational cost. This issue can be further amplified in real-time risk control, where model iteration largely relies on historical review/penalty labels, potentially forming a closed-loop supervision that reinforces false positives over successive retraining cycles. Existing noise-handling methods (e.g., reweighting or filtering) largely treat such errors as random noise and fail to address the root cause—intrinsic feature overlap between scalpers and certain normal users.

We propose **GUARD** (**G**rounded **U**ser-feedback **A**dversarial **R**epresentation **D**isentanglement), a complaint-aware framework that learns risk-predictive representations while being insensitive to complaint-triggering superficial cues. Here, *grounded* means the adversarial supervision is anchored in *complaint-verified* false positives, rather than raw complaints. GUARD defines a *Confusion Domain* from these verified cases and uses it as direct supervision for a GRL-based adversarial objective, encouraging the encoder to be invariant to Confusion-Domain membership while remaining predictive of scalper risk. The model is trained in a multi-task manner with a primary risk head (reliable enforcement labels) and an adversarial confusion head. To mitigate the scarcity and bias of verified complaints, we expand the Confusion Domain via MC Dropout uncertainty sampling, mining potential false-positive candidates from a large pool of processed candidate orders, while filtering out high-confidence scalpers using existing high-precision blacklist rules to reduce contamination.

We evaluate GUARD on a large-scale e-commerce promotion platform. In a 14-day online A/B test with operating thresholds explicitly *calibrated to match the incumbent recall*—ensuring that any precision gain cannot be attributed to a trivially shifted decision boundary—GUARD improves precision by +8.9 points and reduces the complaint rate by 13.5%, while subsidy loss is statistically unchanged at the 5% level. GUARD is deployed in production now and serves millions of daily orders.

## 1 Introduction

Large-scale e-commerce promotions such as a large-scale e-commerce promotion platform have become a core mechanism for user acquisition and retention. Their scale and subsidy intensity, however, also attract professional scalpers who systematically exploit platform rules for profit, undermining promotion fairness and eroding user trust. As a result, accurate and stable scalper detection is mission-critical for both platform safety and user experience. Such promotions often feature deep subsidies that create immediate arbitrage opportunities, incentivizing coordinated scalper behaviors. Figure 1 illustrates a typical subsidy deal and the corresponding secondary-market arbitrage described in scalper group chats.

A dominant difficulty in this setting is not merely the sophistication of scalpers, but an endemic form of *instance-dependent* label noise arising from behavior overlap between malicious and benign users Song et al.

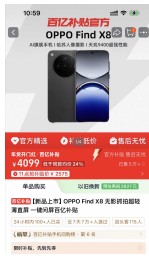 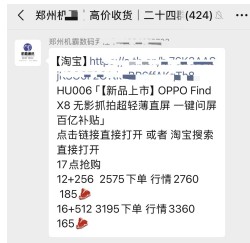

(a) A deep-subsidy deal (OPPO Find X8): 4,099 RMB original vs. 2,575 RMB during the 11:00 flash sale.

(b) Anonymized scalper chat indicating arbitrage: 2,575 RMB order vs. 2,760 RMB buyback (185 RMB margin).

Figure 1: **Motivation: deep subsidies create arbitrage incentives and coordinated scalper actions.** All identifiers (e.g., group names, avatars, phone numbers, links, and order details) are masked for privacy.

(2022); Xiao et al. (2015). Benign users—e.g., fan club organizers coordinating group purchases, families buying for multiple households, or small resellers with legitimate intent—may exhibit patterns that resemble scalping: ordering exactly at event start times, repeatedly purchasing *high-subsidy* items, or shipping to non-habitual/multiple addresses. These behaviors create a practical *confusion zone* where intent is ambiguous given observable features.

Consequently, due to limited model generalization and scalpers' attempts to mimic normal purchasing behaviors, it is inevitable that some of borderline users experience degraded shopping experience, triggering cancellations and appeals. In production, such errors are not only evaluation artifacts; they translate into measurable operational cost and lasting reputation damage. Moreover, deep neural networks can easily overfit spurious correlations in noisy supervision Arpit et al. (2017); Zhang et al. (2016), making the issue persistent even under standard regularization. Figure 2 provides a concrete view of this confusion zone (plotted on a stratified random sample for readability at production scale). Here, label=0 indicates appeal-verified false positives, label=1 confirmed scalpers, and label=2 normal benign users. On the platform subsidy amount, false positives (label=0) exhibit substantial overlap with scalpers (label=1), and the ambiguity persists in the joint space with purchase hour. This suggests that many false positives are *mechanism-driven*—they arise from semantically meaningful behaviors induced by promotion rules—rather than isolated outliers or random label flips.

**Limitations of passive noisy-label learning.** Learning with noisy labels has been extensively studied, with approaches including robust losses, loss correction, and sample selection (e.g., co-teaching and small-loss filtering) Frénay & Verleysen (2013); Han et al. (2018); Patrini et al. (2017); Nigam et al. (2020). While these methods are effective for approximately random or class-conditional noise, they are less suitable when noise is *mechanism-driven*: false positives concentrate on specific, semantically meaningful behaviors that resemble scalping. In our setting, misclassified benign users are not arbitrary outliers to be filtered away; they are evidence that the model has entangled true risk factors with complaint-triggering superficial cues. Treating them as mere noise to down-weight or discard can reduce training signal and does not directly prevent the model from repeatedly relying on the same confusing cues.

**From passive correction to active disentanglement.** To achieve robust scalper detection, the representation should preserve *risk-predictive* factors while becoming invariant to *confusion-inducing* factors. This goal aligns with disentangled representation learning Bengio et al. (2013); Liu et al. (2022), but typical DRL is unsupervised and not tailored to high-stakes classification where explicit post-decision feedback exists. Meanwhile, adversarial learning offers a practical mechanism to enforce invariances, commonly used in domain adaptation via Gradient Reversal Layers (GRL) Ganin & Lempitsky (2015); Ganin et al. (2016). We observe that in promotion risk control, the key "domain" is not an acquisition domain (source vs. target), but a *confusion mechanism* revealed by user appeals and complaints. This motivates us to elevate adversarial training from distribution alignment to *feedback-grounded* representation disentanglement.

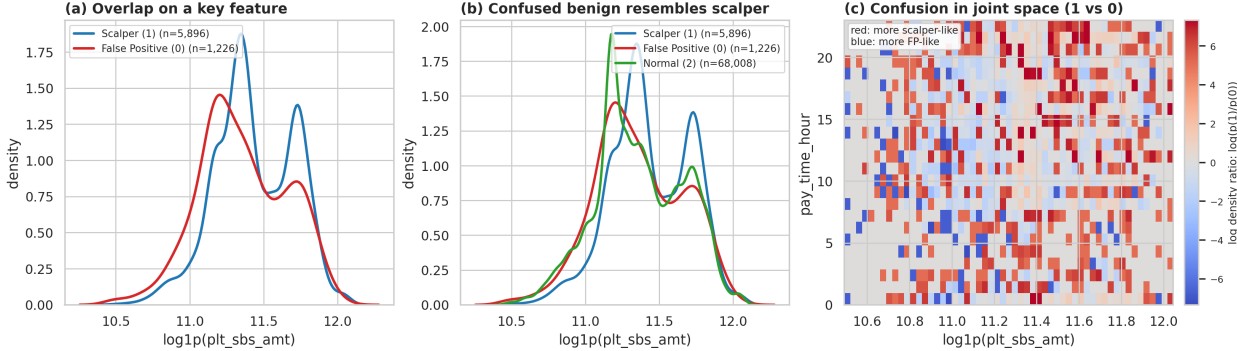

Figure 2: **Mechanism-driven confusion zone in subsidy promotions (sampled for visualization).** Complaint-verified false positives (label=0) exhibit strong distributional overlap with scalpers (label=1) on a key promotion-related feature (platform subsidy amount), and the overlap persists in joint feature space with purchase hour, indicating instance-dependent noise rather than random label flips. *Quantitative overlap.* On the platform subsidy amount marginal, the 1-D total variation distance between the verified-FP distribution and the scalper distribution is approximately 0.18 (95% bootstrap CI [0.16, 0.21]), the symmetrized KL divergence is 0.27, and the 1-Wasserstein distance is 0.09 (normalized). For the benign-vs-scalper reference pair, the same TV statistic is 0.71. The verified-FP–vs–scalper TV is therefore roughly 4× smaller than the benign-vs-scalper TV, confirming that confusion is structural rather than incidental. We plot a stratified random sample from each group for readability at production scale.

**GUARD: grounded user-feedback adversarial representation disentanglement.** We propose **GUARD**, a framework that learns risk-predictive representations while being insensitive to complaint-triggering superficial cues. Here, *grounded* means the adversarial supervision is anchored in *complaint-verified* false positives (appeals confirmed as benign after review), rather than treating raw complaints as ground-truth labels. GUARD explicitly constructs a *Confusion Domain* from these verified cases and trains a shared encoder with two heads: (i) a primary risk classifier optimized on reliable enforcement labels, and (ii) an adversarial confusion classifier that predicts Confusion-Domain membership. A GRL updates the encoder to be predictive for scalper risk while invariant to the confusion mechanism, thereby reducing false positives without sacrificing core risk signals. Because verified complaints are naturally scarce and biased, we further expand the Confusion Domain with MC Dropout uncertainty sampling, mining decision-unstable (high-uncertainty) orders that are likely to be misclassified from a large pool of processed candidate orders.

**Production impact.** GUARD is evaluated and deployed in a large-scale e-commerce promotion platform. In a 14-day online A/B test, GUARD improves precision by +8.9 points at matched recall. Importantly, with the platform subsidy-loss level held constant, GUARD reduces the complaint rate by 13.5%, improving user experience without increasing subsidy leakage. GUARD is deployed in production and serves millions of daily orders.

**Contributions.** Our contributions are threefold:

- We identify a mechanism-driven, instance-dependent noisy-label failure mode in promotion risk control and argue for *active* disentanglement rather than passive noise suppression.

- We propose **GUARD**, a feedback-grounded adversarial disentanglement framework that uses complaint-verified false positives to define a *Confusion Domain* and enforce invariance via a GRL-based multi-task architecture.

- We introduce an uncertainty-based Confusion-Domain expansion strategy using MC Dropout and demonstrate substantial real-world gains in both model metrics (precision) and business metrics (complaint rate) under a fixed subsidy-loss constraint.

## 2 Related Work

### 2.1 Scalper and Fraud Detection

Scalper detection Wu et al. (2018) in large-scale e-commerce promotions can be viewed as a specialized form of fraud and abuse detection, where adversaries exploit platform mechanisms (e.g., deep subsidies and flash-sale timing) for arbitrage while camouflaging as benign users. Industrial systems typically rely on rich heterogeneous signals—user profiles, devices, addresses, behavior sequences, and order attributes—and increasingly adopt deep learning and graph-based modeling Dou et al. (2020) to capture complex cross-entity interactions. For example, graph neural networks have been applied to fraud detection but may suffer from inconsistency issues in practice due to graph construction and distribution shift Liu et al. (2020b). Recent benchmark efforts also revisit supervised graph anomaly detection settings and highlight the gap between academic benchmarks and industrial requirements Tang et al. (2023). Beyond homophilous assumptions, heterophily-aware perspectives further improve graph anomaly detection robustness Gao et al. (2023); Shao et al. (2025). These graph-based studies mainly emphasize relational inductive bias and representation capacity, but they do not directly address a key pain point in promotion risk control: *post-decision feedback* (appeals/complaints) that systematically reveals false-positive mechanisms. Different from conventional fraud detection where malicious patterns can often be separated by strong signals, scalper detection in promotion scenarios exhibits a prominent *mechanism-driven confusion zone*: certain benign subgroups (e.g., group purchase organizers and multi-household buyers) naturally share behavioral signatures with scalpers. Such overlap yields systematic false positives and potential closed-loop supervision issues, where historical review/penalty labels may reinforce false positives over successive retraining cycles. This setting motivates methods that explicitly mitigate confusion mechanisms exposed by user feedback, rather than only optimizing global accuracy.

### 2.2 Learning with Noisy Labels and Instance-Dependent Noise

Learning under noisy supervision has been extensively studied, from early surveys on label-noise classification Frénay & Verleysen (2013); Nigam et al. (2020); Menon et al. (2015); Deng et al. (2025) to recent comprehensive reviews of noisy-label learning with deep neural networks Song et al. (2022). A common line of work assumes class-conditional or approximately random label noise and proposes robust objectives or correction strategies. Loss correction methods estimate noise transitions and adjust the training loss accordingly Patrini et al. (2017), while other approaches study learning from massive noisy labels in recognition tasks Xiao et al. (2015). Sample selection and peer-learning methods, such as Co-teaching Han et al. (2018), train two networks to exchange "small-loss" instances to reduce the influence of noisy labels. More recent co-learning/co-training variants further improve robustness via asymmetric updates and complementary training signals without relying on explicit noise priors, e.g., CA2C Sheng et al. (2025). DivideMix Li et al. (2020) combines GMM-based sample division with semi-supervised MixMatch training, achieving strong results under both symmetric and asymmetric noise on image benchmarks. Early-Learning Regularization (ELR) Liu et al. (2020a) prevents memorization of noisy labels by adding a regularization term that encourages predictions to remain consistent with the model's early-stage outputs.

A central challenge is that deep networks can memorize corrupted supervision Zhang et al. (2016); Arpit et al. (2017), and this effect becomes more severe under *instance-dependent noise* Nguyen et al. (2024); Liao et al. (2025); Garg et al. (2023) where the noise correlates with features. In promotion risk control, noisy supervision is often *mechanism-driven*: false positives concentrate on semantically meaningful patterns induced by platform rules (e.g., on-the-hour purchasing and atypical shipping addresses). In such cases, simply filtering or reweighting samples may either (i) discard informative hard examples or (ii) fail to prevent the model from repeatedly relying on the same spurious but highly predictive cues. This differs from settings where noise can be treated as i.i.d. perturbations around clean labels. To better handle complex noise, recent work has moved toward more adaptive and structured noisy-label learning. Twin Contrastive Learning strengthens robustness by leveraging contrastive objectives under noisy supervision Huang et al. (2023). Progressive sample selection frameworks further combine curriculum-style filtering with representation regularization; for example, PSSCL integrates contrastive loss to gradually refine the training set under noisy labels Zhang et al. (2025). Dynamic instance-dependent selection and correction frameworks such as DISC

aim to jointly identify corrupted labels and correct them during training Li et al. (2023). Decoupled meta label purifier style frameworks separate label purification from predictor learning to reduce error reinforcement and improve stability under heavy noise Tu et al. (2023). Hard-sample mining has also been studied through meta-learning signals; for instance, meta-learning dynamic center distance uses dynamically learned class centers to mine hard examples and improve robustness under label noise Mu et al. (2025). Beyond loss-based filtering, sequence-modeling and neighborhood-based denoising strategies have been explored, e.g., learning with noisy labels via Mamba-style modeling combined with entropy-guided $k$NN mechanisms to refine pseudo labels or stabilize representations Wang et al. (2025). We further relate to confidence-based Positive–Unlabeled (PU) learning under instance-dependent label noise, which studies biased contamination in the unlabeled pool and uses confidence modeling to mitigate estimation bias Tang et al. (2025). Despite their effectiveness, these approaches mainly aim to recover clean labels, identify a trusted subset, or estimate unbiased risk under noise. Our problem has an additional structure: we have *post-decision feedback* indicating that a specific subset of benign users is repeatedly misclassified due to feature overlap. Instead of only denoising labels, we treat complaint-verified false positives as semantic feedback that exposes the *false-positive mechanism*. GUARD leverages this structure by defining a Confusion Domain and explicitly enforcing invariance to Confusion-Domain membership, thereby targeting the root cause of repeated false positives.

## 2.3 Adversarial Learning and Disentangled Representation Learning

Representation learning aims to discover informative features that support downstream tasks and generalize beyond training data Bengio et al. (2013). Disentangled representation learning (DRL) further seeks to separate underlying factors of variation, improving interpretability, robustness, and controllability Wang et al. (2024); Liu et al. (2022). Beyond classical unsupervised DRL, adversarial objectives provide a practical mechanism to enforce invariances and disentangle nuisance factors. Adversarial training is widely known for adversarial-example robustness Goodfellow et al. (2014); Madry et al. (2017), but it is also a standard tool for *domain invariance* through gradient reversal and domain discriminators. Adversarially enforced disentanglement has also been explored in generative modeling, e.g., adversarial disentangling variational autoencoders Silva & Farias (2025), highlighting that adversarial signals can separate task-relevant and nuisance factors when an appropriate supervision signal exists.

Our work is most closely related to domain-adversarial training (DANN) Ganin & Lempitsky (2015); Ganin et al. (2016), but differs in both problem framing and supervision. In promotion risk control, the key "domain" is not a source/target acquisition shift. Instead, it corresponds to a *confusion mechanism* revealed by post-decision user feedback. Naively treating complaint samples as a target domain and aligning them can be counterproductive, because complaint data is not a distribution to match; it is a *highly biased slice* of the population, enriched with false positives and influenced by user behavior and complaint processes. GUARD therefore uses *grounded* adversarial supervision anchored in *complaint-verified* false positives, rather than raw complaints, to reduce label ambiguity and avoid suppressing truly risk-predictive factors.

We implement the adversarial objective via a Gradient Reversal Layer (GRL) Ganin & Lempitsky (2015); Ganin et al. (2016). Compared with more recent disentanglement paradigms based on generative modeling (e.g., VAE-style objectives Higgins et al. (2017); Chen et al. (2018); Mathieu et al. (2019)) or contrastive learning Mo et al. (2023); Li et al. (2021), GRL offers a better fit for ultra-large-scale, latency-sensitive risk control systems: it introduces no reconstruction branch or large-batch contrastive training machinery, requires only an auxiliary confusion head, and adds negligible overhead to training and inference. More importantly, GRL directly exploits the supervision signal available in our setting—Confusion-Domain membership from complaint-verified false positives—to *remove a specific nuisance factor* (complaint-triggering superficial cues) while the primary risk head preserves scalper-predictive information.

Moreover, verified complaints are naturally scarce and biased, which limits direct adversarial supervision. To mitigate this, we expand the Confusion Domain via MC Dropout uncertainty sampling Gal & Ghahramani (2016); Lewis (1995): mining decision-unstable reviewed orders that are more likely to lie in the confusion zone, while filtering out high-confidence scalpers using high-precision blacklist rules to reduce contamination. This differs from generic active learning Settles (2009); Ren et al. (2021); Li et al. (2024), as the objective is not to maximize label efficiency but to increase coverage of the confusion mechanism for representation

invariance. Overall, GUARD re-purposes adversarial invariance from domain alignment to *feedback-grounded disentanglement* under mechanism-driven, instance-dependent noise.

**Why these components, in this order: contrast with DANN and naive negative augmentation.** Although GUARD reuses two well-known building blocks—a Gradient Reversal Layer and MC Dropout uncertainty—its contribution is not the components themselves, but *what supervision signal they consume and what they are asked to remove.* Two contrasts make this concrete. *(i) vs. DANN.* DANN aligns a source and a target distribution under the assumption that both are valid samples from a distribution one wants the model to behave well on. In our setting, the "target" would be the complaint-verified slice $\mathcal{D}_c$, but $\mathcal{D}_c$ is *not* a distribution to match: it is a deliberately biased, post-decision feedback slice that exposes a specific failure mechanism. Aligning to it would preserve, not remove, the confounding cues. GUARD therefore uses the GRL to enforce *invariance* to membership in this biased slice rather than to align with it— a different optimization target that yields a different decision surface (Table 1: DANN 82.3% vs. GUARD 89.7% precision). *(ii) vs. DNN+Conf. Neg.* Adding $\mathcal{D}_c$ as additional negatives only adjusts the decision boundary; the encoder still maps confused benign users and scalpers to nearby regions in latent space, which is exactly the failure mode our t-SNE plots show (Figure 8). The empirical evidence is consistent: this naive use of the same data yields only +2.3% precision (Table 1). GUARD instead reshapes the geometry of the representation, and Eq. equation 7 ties this geometric change to an upper bound on FPR-B, with the two terms attacked respectively by the GRL ($\mathrm{TV}(P_1, P_0)$) and by MC Dropout mining ($\epsilon$). The components are therefore not interchangeable; each addresses a separate term in the same bound.

# 3    The GUARD Framework

**Core intuition.**    Before presenting the formal framework, we provide the key intuition behind GUARD's design. A standard classifier trained on noisy historical labels learns two types of signal: (1) *genuinely risk-predictive factors* such as multi-account address aggregation and coordinated purchase patterns, and (2) *superficial correlates* that co-occur with scalping in training data but also appear among confused benign users, such as purchasing at promotion start times or buying high-subsidy items. The complaint-verified false positives (our Confusion Domain) are precisely the subpopulation that exhibits type-(2) features without type-(1). By training the encoder so that its representations *cannot* distinguish Confusion-Domain samples from regular ones, we effectively remove type-(2) signals from the latent space, forcing the risk head to rely solely on type-(1) genuinely discriminative factors.

This follows the same logic as removing domain-specific nuisance features in domain adaptation Ganin et al. (2016); Ben-David et al. (2010), with one crucial distinction: our "domain" is not an acquisition source or deployment environment, but a *false-positive mechanism* revealed by verified user feedback. The adversarial confusion head acts as a probe for type-(2) features; the Gradient Reversal Layer removes exactly the information this probe exploits, while the risk head preserves type-(1) signals through its own supervised objective. Thus, the two heads perform complementary roles—the risk head retains discriminative power, while the confusion head sculpts the representation to discard confounding patterns.

## 3.1    Problem Formulation

Let an order be represented by a feature vector $x \in \mathcal{X}$. The true, unobservable label is $y \in \{0, 1\}$, where $y = 1$ denotes a scalper and $y = 0$ denotes a benign user. Our training data consists of two main parts:

- **Reliably labeled data.** A set of reliably labeled data $\mathcal{D}_r = \{(x_i, y_i)\}$. Positive labels ($y = 1$) come from a high-precision, historically accumulated scalper blacklist, where cases have clear supporting evidence and are manually verified by risk-control experts (e.g., strong address aggregation patterns and other corroborating traces discovered during investigations). Negative labels ($y = 0$) are sampled from regular traffic following the standard production labeling pipeline.

- **Complaint-verified false positives.** A set of complaint-verified false positives $\mathcal{D}_c = \{x_j\}$, where each $x_j$ is an order that was initially flagged by the online risk system but later confirmed to be benign through a verified complaint workflow. Specifically, a case enters $\mathcal{D}_c$ only after a multi-stage

process: user complaint $\rightarrow$ human review $\rightarrow$ secondary confirmation, and only then is it fed back to the training data. These samples form our initial "Confusion Domain." **Definition (explicit vs. implicit confusion samples).** We refer to $\mathcal{D}_c$ as *explicit confusion samples* because each case is a verified false positive returned by the complaint workflow. In addition, we will mine a set of *implicit confusion candidates* $\mathcal{S}$ (Sec. 3.5) from the reviewed unlabeled pool $\mathcal{U}$ using MC Dropout uncertainty. Both $\mathcal{D}_c$ and $\mathcal{S}$ are used *only* to supervise the confusion classifier $G_d$ (domain label $d = 1$), while the risk classifier $G_y$ is trained on reliable labels from $\mathcal{D}_r$ (Sec. 3.3).

The core challenge is that there exists a subset of benign users whose features are statistically similar to those of scalpers, leading a standard classifier $f(x)$ to misclassify them. Our goal is to learn a robust classifier that minimizes misclassification, especially for these confusing benign users.

## 3.2 Framework Overview

As illustrated in Figure 3, GUARD consists of three main components:

1. **Shared Feature Encoder ($G_f$):** A deep neural network that maps the input feature vector $x$ to a latent representation $f = G_f(x; \theta_f)$.

2. **Risk Classifier ($G_y$):** A classifier head that takes $f$ and predicts the probability of the order being from a scalper, $p(y = 1|x) = G_y(f; \theta_y)$.

3. **Adversarial Confusion Classifier ($G_d$):** Another classifier head, preceded by a Gradient Reversal Layer (GRL), that predicts whether an order belongs to the Confusion Domain. Let $d \in \{0, 1\}$ be the domain label ($d = 1$ for Confusion Domain, $d = 0$ for regular training data). This head predicts $p(d = 1|x) = G_d(f; \theta_d)$.

The framework operates in a multi-task learning setting. The risk classifier $G_y$ is trained to minimize classification error on reliable data. Simultaneously, the confusion classifier $G_d$ is trained to distinguish confusion samples, but the GRL ensures that the shared encoder $G_f$ receives a reversed gradient, forcing it to learn features that are *indistinguishable* to $G_d$.

## 3.3 Adversarial Disentanglement Module

The core of GUARD is the interplay between the risk and confusion classifiers. The loss for the risk classifier, $L_y$, is a standard binary cross-entropy loss computed on the reliable dataset $\mathcal{D}_r$:

$$L_y(\theta_f, \theta_y) = -\mathbb{E}_{(x,y)\sim\mathcal{D}_r}\Big[y\log(G_y(G_f(x))) + (1-y)\log(1 - G_y(G_f(x)))\Big]. \tag{1}$$

The loss for the confusion classifier, $L_d$, is also a binary cross-entropy loss, trained to distinguish samples from the **explicit confusion set** $\mathcal{D}_c$ from samples in $\mathcal{D}_r$:

$$\mathcal{L}_d(\theta_f, \theta_d) = -\mathbb{E}_{x\sim\mathcal{D}_r}\big[\log\big(1 - G_d(G_f(x))\big)\big] - \mathbb{E}_{x\sim\mathcal{D}_c}\left[\log G_d(G_f(x))\right]. \tag{2}$$

The GRL does not change the forward pass but reverses the gradient during backpropagation. The parameters $\theta_f$, $\theta_y$, and $\theta_d$ are updated according to the following objectives:

$$(\hat{\theta}_f, \hat{\theta}_y) = \arg\min_{\theta_f, \theta_y}\left(L_y - \lambda L_d\right), \tag{3}$$

$$\hat{\theta}_d = \arg\min_{\theta_d} L_d, \tag{4}$$

where $\lambda$ controls the strength of confusion invariance. By minimizing $L_y$ and maximizing $L_d$ with respect to the encoder parameters $\theta_f$, the encoder is encouraged to learn representations that are predictive for the risk task but invariant to the confusing characteristics captured by $\mathcal{D}_c$.

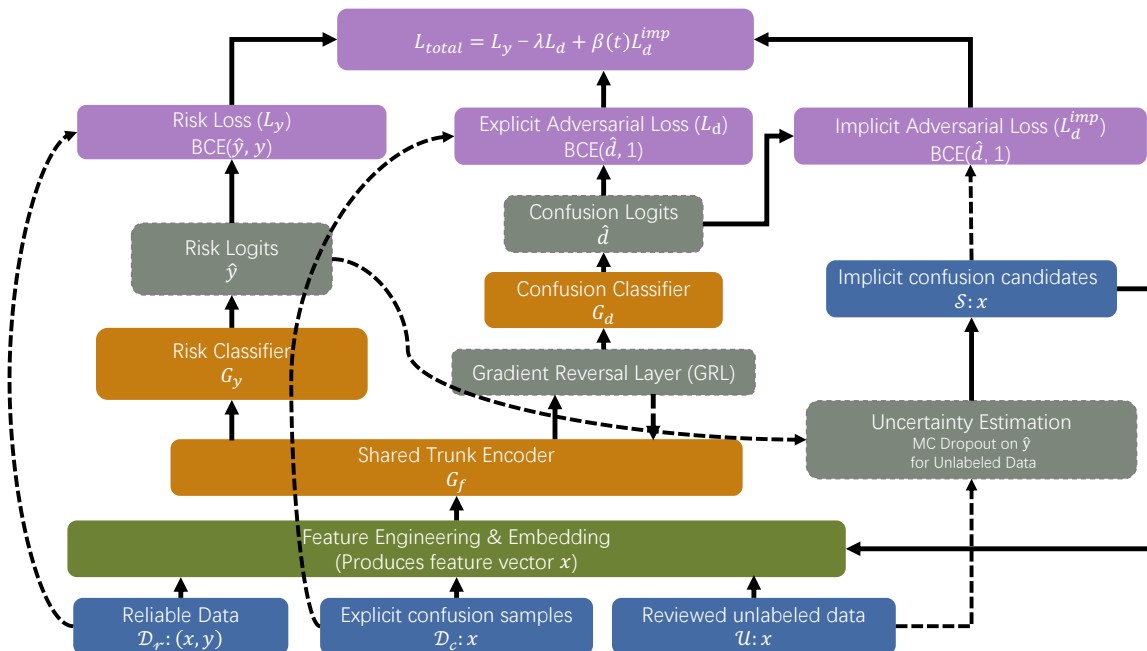

Figure 3: The architecture of the GUARD framework. Raw order attributes are transformed into an input feature vector $x$ via feature engineering and embeddings, and then encoded by a shared trunk encoder $G_f$. The representation is fed into two heads: (i) a risk classifier $G_y$ that outputs risk prediction $\hat{y} = G_y(G_f(x))$ and is trained on reliable labels from $\mathcal{D}_r$ with the risk loss $L_y$; and (ii) a confusion (domain) classifier $G_d$ that outputs domain prediction $\hat{d} = G_d(G_f(x))$ and is connected to $G_f$ through a Gradient Reversal Layer (GRL) to enforce confusion-invariant features. The confusion head is supervised by (a) explicit confusion samples $\mathcal{D}_c$ (complaint-verified false positives) and (b) implicit confusion candidates $\mathcal{S}$ mined from the reviewed unlabeled pool $\mathcal{U}$ via MC Dropout uncertainty. Mined candidates contribute an additional annealed loss term $\beta(t)\, L_d^{imp}$.

### 3.4 Theoretical Analysis

We provide a formal analysis showing that GUARD's adversarial objective directly reduces the false positive rate on confusion-prone users, and that uncertainty-based mining complements this by reducing coverage gaps.

**Setup.** Let $z = G_f(x)$ be the shared representation. Let $D \in \{0, 1\}$ indicate whether a sample belongs to the Confusion Domain used for adversarial supervision ($D = 1$: complaint-verified false positives plus mined candidates; $D = 0$: regular reliable data). We evaluate false positives on a confusion-prone benign cohort $B = 1$ (Sec. 4.1), constructed by a conjunction of interpretable rules and validated by expert sampling. Empirically, verified complaint cases largely fall into $B = 1$, so $D = 1$ provides a high-precision subset signal for the confusion zone.

**GRL reduces domain distinguishability in $z$.** Let $\mathcal{H}_d$ be the hypothesis class of the confusion classifier. Define the $\mathcal{H}_d$-divergence between two distributions over $z$ as

$$d_{\mathcal{H}_d}(P, Q) = 2 \sup_{h \in \mathcal{H}_d} \left| \Pr_{z \sim P}[h(z) = 1] - \Pr_{z \sim Q}[h(z) = 1] \right|. \tag{5}$$

As established in domain adaptation theory Ganin et al. (2016); Ben-David et al. (2010), maximizing the confusion loss w.r.t. the encoder (via GRL) reduces the ability of any $h \in \mathcal{H}_d$ to discriminate $D = 1$ from $D = 0$, thereby decreasing $d_{\mathcal{H}_d}(p(z \mid D = 1), p(z \mid D = 0))$.

**Motivating observation on FPR-B.** We give a simple observation connecting adversarial invariance to false positive reduction. Consider a threshold classifier $\hat{y} = \mathbb{I}[s(z) \geq t]$ where $s : \mathcal{Z} \to \mathbb{R}$ is a scoring function over representations. Define FPR-B$(t) = \Pr(s(Z) \geq t \mid Y = 0, B = 1)$ as the false positive rate on the confusion-prone benign cohort. Let $P_1 = p(z \mid Y = 0, D = 1)$, $P_0 = p(z \mid Y = 0, D = 0)$, and define the coverage gap

$$\epsilon = \text{TV}(p(z \mid Y = 0, B = 1),\, P_1). \tag{6}$$

By the triangle inequality for total variation distance, for any threshold $t$,

$$\left| \text{FPR-B}(t) - \Pr(s(Z) \geq t \mid Y = 0, D = 0) \right| \leq \text{TV}(P_1, P_0) + \epsilon. \tag{7}$$

**Interpretation.** Eq. equation 7 decomposes the excess FPR-B into two controllable terms:

- $\text{TV}(P_1, P_0)$: the *domain discrepancy* in representation space between confusion-domain and regular benign users. GRL training directly minimizes this term—as the confusion head approaches random guessing (empirically verified in Figure 6), the $\mathcal{H}_d$-divergence (and hence TV) decreases.

- $\epsilon$: the *coverage gap* between the evaluation cohort ($B = 1$) and the training Confusion Domain ($D = 1$). Uncertainty-based mining (Sec. 3.5) expands the Confusion Domain to cover more of the confusion zone, reducing $\epsilon$.

Together, adversarial invariance and uncertainty mining attack both terms, providing a principled mechanism for FPR-B reduction. Note that this bound is agnostic to the risk head's discriminative quality; the risk head is preserved by its own supervised objective on reliable labels, and the GRL ensures gradients from the confusion head do not improve confusion-domain discrimination in the encoder.

### 3.5 Augmenting the Confusion Domain

User complaint data $\mathcal{D}_c$ is often scarce. We therefore mine implicit confusion candidates $\mathcal{S}$ from the reviewed unlabeled pool $\mathcal{U}$. In training the confusion classifier $G_d$, we assign *domain label* $d = 1$ to both $\mathcal{D}_c$ and $\mathcal{S}$, and assign $d = 0$ to samples from $\mathcal{D}_r$. Note that $\mathcal{S}$ is *not* used as negative labels for the risk task; it only strengthens the adversarial confusion supervision. To address this, we propose an uncertainty-guided augmentation strategy using Monte Carlo (MC) Dropout Gal & Ghahramani (2016). We choose MC Dropout over alternative uncertainty methods for three reasons: (i) it incurs near-zero additional cost by reusing the existing dropout layers from training—no auxiliary models or ensembles are needed, which is critical given our production system processes millions of daily orders; (ii) it provides *epistemic* uncertainty estimates that are semantically aligned with our goal: high predictive variance indicates decision instability, which is precisely the hallmark of confusion-zone samples—genuine scalpers and clearly benign users both receive stable predictions, while borderline cases oscillate; (iii) compared with ensemble-based methods (which require $N\times$ parameters and memory) or evidential deep learning (which requires architectural changes), MC Dropout integrates seamlessly into our existing training and serving infrastructure.

For an unlabeled order $x_u$ (sourced from a production *reviewed pool*, i.e., orders initially flagged by the online model), we perform $T$ stochastic forward passes with dropout enabled, obtaining predictions $\{p_t(y = 1|x_u)\}_{t=1}^{T}$.

We measure uncertainty by the variance:

$$\text{Uncertainty}(x_u) = \text{Var}(\{p_t(y = 1|x_u)\}_{t=1}^{T}). \tag{8}$$

**Selecting confusion candidates (code-aligned).** Variance alone tends to select samples near the decision boundary, but we additionally filter out *overly confident positives* to avoid mining clear scalpers. Concretely, let $\mu(x_u) = \frac{1}{T}\sum_{t=1}^{T} p_t(y = 1|x_u)$ be the mean prediction. We keep samples with $\mu(x_u) < \tau$ (e.g., $\tau = 0.9$ in our implementation), and then select Top-$K$ by uncertainty:

$$\mathcal{S} = \text{TopK}_{x_u \in \mathcal{U}:\mu(x_u) < \tau}\ \text{Uncertainty}(x_u). \tag{9}$$

These high-uncertainty, not-overconfident reviewed orders are more likely to correspond to confusion-zone benign users. We add $\mathcal{S}$ to our Confusion Domain for the next training round.

**Simulated-annealing style weighting for mined samples.** The mined set $\mathcal{S}$ can be noisy in early training because the risk predictor is not yet calibrated. Instead of annealing the main adversarial term, we anneal the *additional loss contributed by mined samples*. We define an implicit confusion loss:

$$L_d^{imp}(\theta_f, \theta_d) = -\mathbb{E}_{x \sim \mathcal{S}} \left[ \log G_d(G_f(x)) \right], \tag{10}$$

and optimize the total objective:

$$L_{\text{total}} = L_y - \lambda L_d + \beta(t) \, L_d^{imp}, \tag{11}$$

where $\beta(t)$ increases from $\beta_{\min}$ to $\beta_{\max}$ during the first $T_{\text{anneal}}$ epochs:

$$\beta(t) = \begin{cases} \beta_{\min} + (\beta_{\max} - \beta_{\min}) \cdot \frac{t}{T_{\text{anneal}}}, & t < T_{\text{anneal}}, \\ \beta_{\max}, & t \geq T_{\text{anneal}}. \end{cases} \tag{12}$$

This linear-annealing schedule down-weights unreliable mined samples at the beginning and progressively enforces invariance to confusion mechanisms as training stabilizes.

## 4 Experiments

### 4.1 Experimental Setup

**Dataset.** We conduct experiments on a large-scale industrial dataset collected from a large-scale e-commerce promotion platform over 10 days. The dataset contains a large number of orders. Each order is represented by a feature vector derived from user profile, device, behavioral signals, and order characteristics.

- **Training Set:** A large collection of historical orders with *reliable labels*, where positives come from a high-precision blacklist accumulated over time and validated via periodic manual spot-checking. The training set is on the order of $10^7$ orders, with a positive (scalper) ratio of approximately 1–3%.

- **Confusion Domain:** A smaller set of orders confirmed as false positives through verified user complaints. $|\mathcal{D}_c|$ is on the order of $10^4$ verified cases over the collection window, after the multi-stage *user complaint $\rightarrow$ human review $\rightarrow$ secondary confirmation* workflow. We do not disclose exact volumes due to commercial sensitivity, but report orders of magnitude to allow assessment of statistical reliability.

- **Unlabeled Pool:** A large pool of reviewed high-risk orders without reliable labels, used for uncertainty-based mining; orders already identified as clearly risky by the same reliable-label sources are removed to avoid trivial positives. $|\mathcal{U}|$ is on the order of $10^5$–$10^6$ reviewed orders.

- **Test Set:** Orders from a subsequent time window, independently manually labeled to provide ground truth for evaluation. The test set is on the order of $10^5$ orders. The confusion-prone benign cohort $B=1$ used for FPR-B (defined below) covers approximately 5–10% of the test set, validated by stratified expert sampling.

**Baselines.** To ensure a fair comparison, we organize baselines by the data they access. Methods in the first group use only reliable labels $\mathcal{D}_r$; those in the second group additionally use the confusion domain $\mathcal{D}_c$ (and optionally the unlabeled pool $\mathcal{U}$), matching or approximating the data available to GUARD.

- **DNN:** A standard MLP classifier trained on $\mathcal{D}_r$ only.

- **LightGBM:** A gradient-boosted decision tree model trained on $\mathcal{D}_r$ as a strong non-neural baseline for tabular risk modeling.

- **Co-teaching** Han et al. (2018): A representative noisy-label learning method trained on $\mathcal{D}_r$. We treat complaint data as noisy negative labels.

---

**Algorithm 1** Training procedure of GUARD

---

**Require:** Reliable dataset $\mathcal{D}_r$, explicit confusion dataset $\mathcal{D}_c$, reviewed unlabeled pool $\mathcal{U}$
**Require:** Adversarial weight $\lambda$, implicit-loss schedule $\beta(t)$
**Require:** Learning rate $\eta$, MC-Dropout passes $T$, confidence threshold $\tau$, Top-$K$ budget $K$

1: Initialize model parameters $\theta_f, \theta_y, \theta_d$
2: **for** each training iteration $t$ **do**
3:     Sample reliable mini-batch $\mathcal{B}_r = \{(x_i, y_i)\}_{i=1}^m \sim \mathcal{D}_r$
4:     Sample explicit confusion mini-batch $\mathcal{B}_c = \{(x_j, d_j)\}_{j=1}^{m'} \sim \mathcal{D}_c$
5:     Sample unlabeled mini-batch $\mathcal{B}_u = \{x_u\}_{u=1}^n \sim \mathcal{U}$
6:     Compute risk prediction on $\mathcal{B}_r$:
$$\hat{y}_i \leftarrow G_y(G_f(x_i))$$
7:     Compute risk loss:
$$L_y \leftarrow \text{BCE}(\hat{y}_i, y_i)$$
8:     Compute confusion prediction on $\mathcal{B}_c$ through GRL:
$$\hat{d}_j \leftarrow G_d(G_f(x_j))$$
9:     Compute explicit adversarial loss:
$$L_d \leftarrow \text{BCE}(\hat{d}_j, d_j)$$
10:     Mine implicit confusion candidates $\mathcal{S}$ from $\mathcal{B}_u$ using MC Dropout
     keep samples with $\mu(x) < \tau$ and select the Top-$K$ by predictive variance
11:     **if** $\mathcal{S}$ is not empty **then**
12:         Compute confusion prediction on mined samples through GRL:
$$\hat{d}_s \leftarrow G_d(G_f(\mathcal{S}))$$
13:         Assign implicit domain labels 1 and compute
$$L_d^{\text{imp}} \leftarrow \text{BCE}(\hat{d}_s, 1)$$
14:     **else**
15:         $L_d^{\text{imp}} \leftarrow 0$
16:     **end if**
17:     Compute total loss:
$$L_{\text{total}} \leftarrow L_y - \lambda L_d + \beta(t) L_d^{\text{imp}}$$
18:     Update $(\theta_f, \theta_y, \theta_d)$ using Adam with learning rate $\eta$
19: **end for**

---

- **DNN + Conf. Neg.:** The same DNN architecture, but trained on $\mathcal{D}_r \cup \mathcal{D}_c$ with complaint-verified false positives explicitly added as negative samples ($y = 0$). This is the most straightforward way to utilize the confusion domain.

- **DNN + Reweight:** The DNN trained on $\mathcal{D}_r$, with upweighted negative samples that are most similar to $\mathcal{D}_c$ (measured by cosine similarity in feature space). This tests whether emphasizing confusion-like negatives suffices.

- **DANN** Ganin & Lempitsky (2015): A domain adaptation method that uses GRL to align features between $\mathcal{D}_r$ and $\mathcal{D}_c$ (treated as "target domain"), but without our explicit disentanglement framing or uncertainty mining.

- **GUARD-no-aug:** An ablation of GUARD without uncertainty sampling augmentation (uses $\mathcal{D}_r + \mathcal{D}_c$ but not $\mathcal{U}$).

**Baseline selection rationale.** Our related work surveys several recent noisy-label methods (e.g., DivideMix Li et al. (2020), ELR Liu et al. (2020a), DISC Li et al. (2023), TCL Huang et al. (2023), PSSCL Zhang et al. (2025)) and graph-based fraud detectors (e.g., Dou et al. (2020); Gao et al. (2023)). We deliberately do not include them as baselines, and this absence itself reflects the research gap that motivates our work.

*(i) Noisy-label methods.* Existing noisy-label methods—including recent advances such as DivideMix, ELR, DISC, TCL, and PSSCL—are designed under the assumption that label noise is *random* (symmetric, asymmetric, or instance-dependent but without structured feedback). Their core mechanisms (contrastive augmentation, GMM-based sample division, loss regularization, label purification, or curriculum-based selection) aim to *recover clean labels* or *down-weight corrupted samples*. However, in our setting the noise is *mechanism-driven*: false positives are not arbitrary corruptions to be filtered, but systematic errors caused by intrinsic feature overlap between scalpers and certain benign users, induced by promotion rules. More critically, none of these methods leverage *post-decision feedback*—the verified complaint signal that directly reveals *why* and *where* the model fails. This is precisely the gap GUARD addresses: rather than treating noise as something to passively suppress, we use complaint-verified false positives as active supervision to disentangle the confusion mechanism at the representation level. To the best of our knowledge, no existing noisy-label method provides this feedback-grounded disentanglement capability, which is a key motivation for this work. Among noisy-label baselines, we include Co-teaching Han et al. (2018) as a representative peer-learning method because its small-loss sample exchange paradigm is the most directly applicable to our tabular setting without requiring image-specific augmentation strategies.

*(ii) Graph-based methods.* Graph neural network approaches require constructing explicit relational graphs (e.g., user–device–address interaction networks), which constitutes a fundamentally different modeling paradigm from tabular feature-based classification. Our production system operates on per-order feature vectors under strict latency constraints ($<50\,\mathrm{ms}$); integrating graph construction and message passing would require substantial infrastructure changes that are orthogonal to the representation disentanglement contribution of this work. We note that GUARD's adversarial disentanglement module is architecture-agnostic and could in principle be combined with graph-based encoders; we leave this integration as future work.

**Metrics.** We evaluate models on Precision, Recall, F1-Score, and AUC. Crucially, we also report the **False Positive Rate on Benign Users (FPR-B)**, which measures the rate of misclassifying users who are behaviorally similar to those in the complaint data. Specifically, we construct a *confusion-prone benign cohort* ($B{=}1$) by taking the conjunction of the following interpretable, white-box risk rules: (i) the order was placed within 10 minutes of a flash-sale start time; (ii) the platform subsidy amount exceeds the 75th percentile of the promotion category; (iii) the shipping address differs from the user's most frequent historical address; and (iv) the user has placed $\geq 3$ orders in the same promotion category within the collection window. Each rule individually captures a behavior that is common among both scalpers and confused benign users; their conjunction identifies users who exhibit multiple scalper-like traits simultaneously. We then have operations experts conduct stratified sampling (approximately 500 cases) to verify that the selected users are indeed benign, achieving a verification agreement rate of $>90\%$ among annotators (Cohen's $\kappa > 0.85$). Empirically, the complaint-verified false positives show a high overlap ($>80\%$) with this cohort, making FPR-B a reliable proxy for measuring false positives in the confusion zone.

### 4.2 Implementation Details and Reproducibility

**Model.** GUARD is implemented as a two-head adversarial network with a shared MLP trunk and a GRL-based confusion head. We use 375 continuous features and 26 discrete categorical features; we embed the discrete features and concatenate them with the continuous features as the model input. The trunk is a 2-layer MLP with hidden size 64 and dropout 0.5, followed by two heads with latent size 32 (risk head for risk prediction and adversarial head for confusion-domain prediction). The GRL weight is set to $\lambda = 0.4$.

**Training.** We train with Adam (lr$=10^{-4}$) for 50 epochs with early stopping (patience$=8$) on a validation split (20% of reliably labeled data). To handle class imbalance in the risk task, we use `BCEWithLogitsLoss` with positive-class weight 5. Each iteration uses mini-batches from three sources: reliable data (batch size 512), explicit confusion samples (256), and reviewed unlabeled data (1024). The total loss is the sum of

Table 1: Main experimental results on the industrial dataset. Best results per column are in **bold**. All methods in the upper block use only reliable labels $\mathcal{D}_r$; methods marked with † additionally use the confusion domain $\mathcal{D}_c$; GUARD further uses the unlabeled pool $\mathcal{U}$ for uncertainty mining.

| Method | Precision (%) | Recall (%) | F1 (%) | FPR-B (%) |
|---|---|---|---|---|
| DNN | 78.5±0.4 | **82.1±0.3** | 80.3±0.3 | 15.6±0.3 |
| LightGBM | 79.1±0.3 | 81.0±0.2 | 80.0±0.2 | 14.3±0.3 |
| Co-teaching | 80.2±0.5 | 79.5±0.4 | 79.8±0.4 | 13.1±0.4 |
| DNN + Conf. Neg.† | 80.8±0.4 | 81.6±0.3 | 81.2±0.3 | 12.4±0.3 |
| DNN + Reweight† | 81.3±0.3 | 81.2±0.3 | 81.3±0.3 | 11.8±0.3 |
| DANN† | 82.3±0.4 | 80.8±0.4 | 81.5±0.4 | 8.6±0.3 |
| GUARD-no-aug† | 87.6±0.4 | 81.5±0.3 | 84.4±0.3 | 5.3±0.2 |
| **GUARD** | **89.7±0.3** | 80.2±0.3 | **84.7±0.3** | **5.1±0.2** |

the risk loss, an explicit adversarial loss (weight $\lambda = 0.4$), and an implicit adversarial loss whose weight is linearly annealed from 0.01 to 0.4 over the first 50 epochs.

**MC Dropout mining.** For each unlabeled batch, we perform MC Dropout with $T = 20$ stochastic forward passes on the risk head, compute predictive variance, filter out overly confident positives by mean prediction ($\mu < \tau$, $\tau = 0.9$), and select Top-$K$ uncertain samples with $K = 16$ as implicit confusion candidates. These candidates are used only for the adversarial confusion supervision.

**Mining quality control.** A potential concern is that the mined set $\mathcal{S}$ may contain genuine scalpers, which would cause the adversarial head to suppress risk-predictive features. We apply two safeguards: (i) a mean-prediction filter ($\mu < \tau = 0.9$) that removes overly confident positives, and (ii) exclusion of orders already identified by the high-precision blacklist used to construct reliable positive labels in $\mathcal{D}_r$. To assess mining quality, we conducted a stratified manual audit on a random sample of 200 mined candidates from $\mathcal{S}$. Domain experts classified each candidate as benign, ambiguous, or clearly scalper. The audit found that approximately 78% of mined candidates were benign or ambiguous (i.e., legitimately in the confusion zone), while approximately 22% exhibited scalper-indicative patterns. This contamination rate is substantially lower than the base rate in the reviewed pool $\mathcal{U}$ (which contains ∼40–50% positives before filtering), confirming that the uncertainty-based selection combined with mean-prediction filtering effectively enriches for confusion-zone samples. Furthermore, the annealing schedule $\beta(t)$ down-weights the contribution of $\mathcal{S}$ during early training when mining quality is lowest, providing additional robustness against contamination.

**Online threshold calibration and significance test.** For online A/B testing, we calibrate GUARD's threshold to match the baseline recall on a calibration set, and report precision and complaint metrics at matched recall. Subsidy loss is monitored on daily aggregates over the 14-day window and compared using a two-sided significance test at the 5% level.

### 4.3 Offline Evaluation

**RQ1: Overall Performance.** Table 1 shows the main comparison. GUARD significantly outperforms all baselines across all metrics, especially in Precision and FPR-B. This demonstrates its superior ability to reduce false positives while maintaining high recall. The 11.2% increase in precision over the standard DNN is a substantial improvement in a real-world risk control system.

**Discussion of recall.** We note that the standard DNN achieves the highest recall (82.1%), which is expected: without invariance constraints, it aggressively classifies any sample exhibiting scalper-correlated features as positive. However, this comes at a 3× higher FPR-B (15.6% vs. 5.1%), translating directly to more user complaints in production. Moreover, simply adding the confusion domain as negative samples (DNN + Conf. Neg.) yields only modest precision gains (+2.3%) while preserving high recall, confirming that naively using extra data without representation-level invariance cannot address the feature entanglement problem. *Crucially, in the online deployment we calibrate operating thresholds to match recall (Sec. 4.5,*

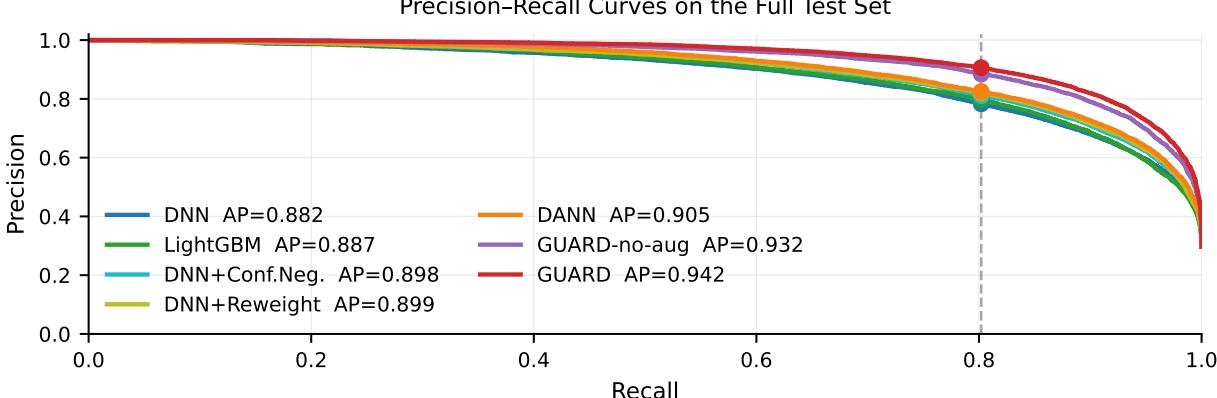

Figure 4: Precision–Recall curves on the full test set. GUARD achieves the best trade-off and the highest AP.

Table 2: Ablation study of GUARD's components.

| Method | Precision(%) | F1-Score(%) |
|---|---|---|
| GUARD | **89.7±0.3** | **84.7±0.3** |
| w/o GRL | 81.5±0.5 | 81.1±0.4 |
| w/o Augmentation | 87.6±0.4 | 84.4±0.3 |

*Table 3), and GUARD still achieves +8.9 precision points under this constraint.* This rules out the most natural alternative explanation—that the offline gains come from a trivially shifted decision boundary—and establishes that the gains stem from a higher-quality representation.

**Precision–Recall Trade-off.** To further compare models under varying decision thresholds, we plot precision–recall curves on the full test set in Figure 4. GUARD achieves the best overall trade-off with the highest average precision (AP=0.942), outperforming strong baselines such as DANN (AP=0.905) and the standard DNN (AP=0.882).

**RQ1': Where do the gains come from? (Confusion cohort vs. Others).** To verify that GUARD mainly reduces false positives in the confusion zone rather than uniformly shifting the decision boundary, we report group-wise FPR under a single global operating point. Specifically, we calibrate one threshold per model on the full test set to match recall (Recall=0.802), and then compute FPR separately on the confusion-prone cohort ($B=1$) and the remaining users ($B=0$). Figure 5 shows that the FPR reduction is concentrated on $B=1$ (15.6%→5.1%), while the change on $B=0$ is minimal (1.21%→1.16%), indicating that GUARD targets the mechanism-driven confusion region without sacrificing overall risk control.

**RQ2: Ablation Study.** We analyze the contribution of each component of GUARD. Table 2 shows that removing either the adversarial disentanglement module (GRL) or the uncertainty sampling augmentation leads to a performance drop, particularly in precision. This validates that both active disentanglement and data augmentation are crucial for GUARD's success.

**RQ2': Does GUARD actually enforce confusion-invariant representations?** Beyond the ablation results, we directly track the discriminability of the Confusion Domain during training. We evaluate the domain (confusion) head on a held-out validation split and report its AUC/accuracy over epochs. As shown in Figure 6, the domain head performance gradually degrades toward random guessing (AUC ≈ 0.5), indicating that the encoder learns representations that are increasingly indistinguishable between confusion samples and regular traffic. Meanwhile, the risk head remains stable, suggesting that invariance is achieved without sacrificing risk-predictive information.

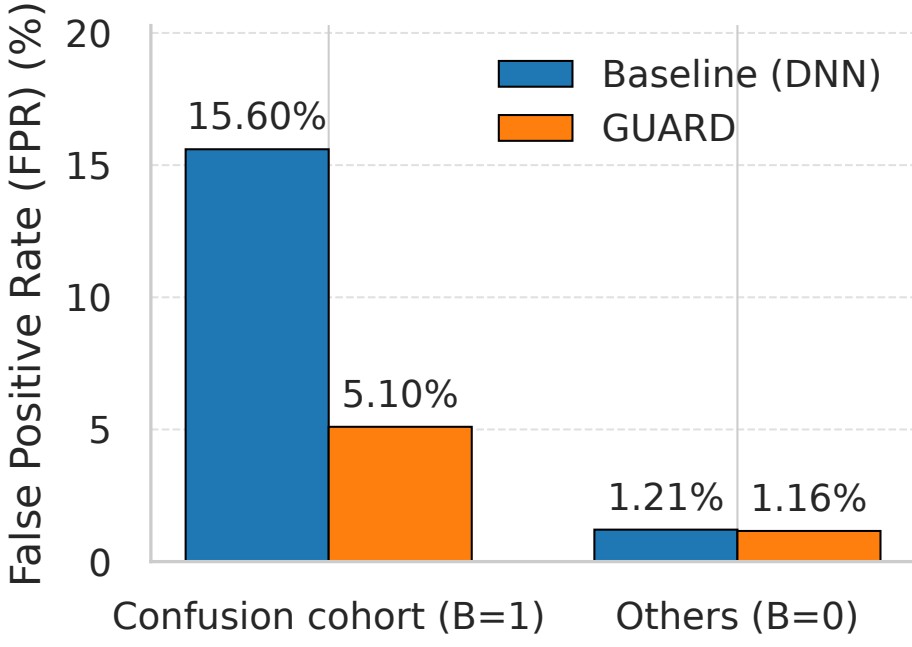

Figure 5: Grouped false positive rates (FPR) at matched recall (Recall=0.802). Thresholds are calibrated on the full test set and applied to both the confusion-prone cohort ($B$=1) and the remaining users ($B$=0). The improvement is concentrated on $B$=1 with minimal impact on $B$=0.

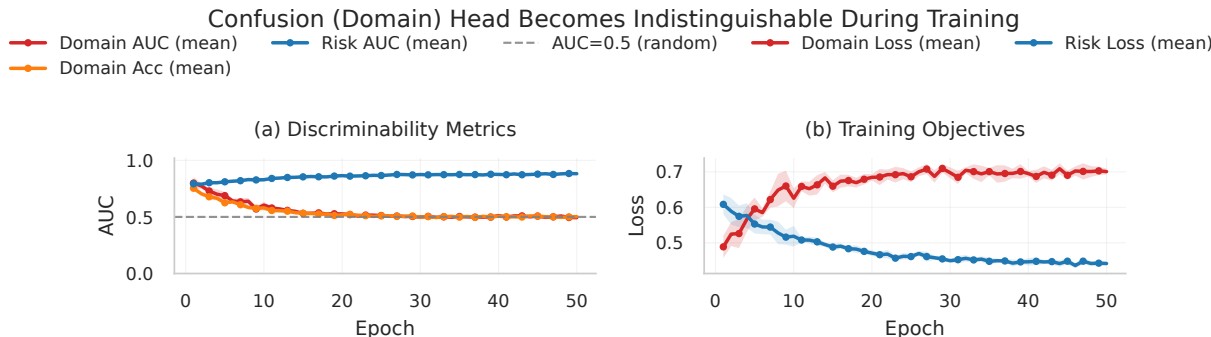

Figure 6: Training dynamics of the confusion (domain) head. Under GRL, the domain head AUC/accuracy on a held-out validation split approaches random guessing (AUC $\approx$ 0.5), indicating effective confusion-invariant representation learning, while the risk head remains stable.

**RQ3: Hyperparameter Sensitivity.** We investigate the effect of the adversarial weight $\lambda$. Figure 7 shows that as $\lambda$ increases, precision improves up to a point ($\lambda = 0.5$) and then slightly degrades, as too much emphasis on disentanglement can harm the primary classification task. This shows the importance of balancing the two objectives.

## 4.4 Qualitative Analysis

**RQ4: Visualization of Disentanglement.** To visually inspect whether GUARD truly learns disentangled representations, we use t-SNE to project the feature vectors from the encoder $G_f$ into a 2D space. Figure

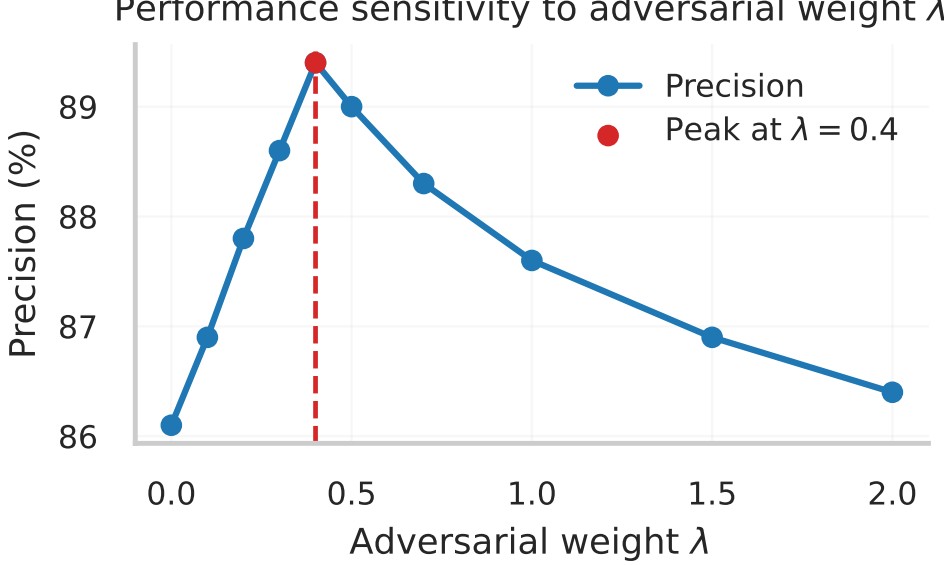

Figure 7: Performance sensitivity to the adversarial weight $\lambda$. Precision peaks around $\lambda = 0.4$.

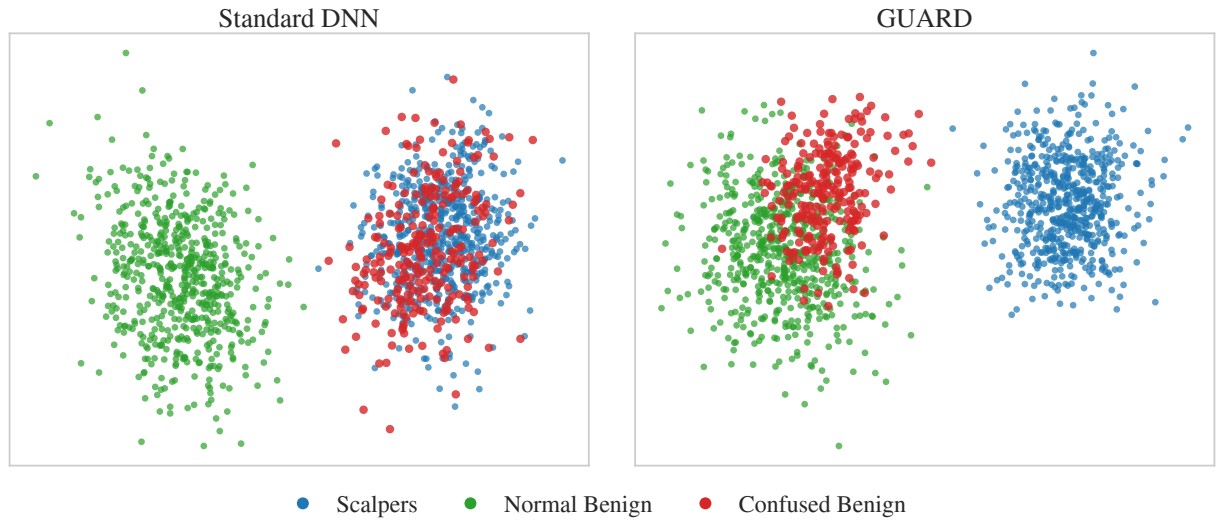

Figure 8: t-SNE visualization of feature representations. Left: Standard DNN. Right: GUARD. (Blue: Scalpers, Green: Normal Benign, Red: Confused Benign). GUARD successfully separates the confused users from the scalper cluster.

8 shows the feature distributions for a standard DNN and for GUARD. In the DNN's feature space, the "Confused Benign" users (from the complaint data) are heavily interspersed with the "Scalper" cluster. In contrast, GUARD successfully pushes the "Confused Benign" users away from the "Scalper" cluster and closer to the "Normal Benign" cluster, providing clear visual evidence of effective feature disentanglement.

Table 3: Online A/B testing results (14-day period) with thresholds calibrated to match recall. Complaint-related and subsidy-loss metrics are reported in normalized form (baseline=1.0) due to data sensitivity. Confidence intervals and $p$-values are computed from $n$=14 daily aggregates using a two-sided test.

| Metric | Baseline (DNN) | GUARD (Ours) | 95% CI ($p$-value) |
|---|---|---|---|
| Precision (matched recall) | 80.1% | 89.0% | $[+7.9, +9.9]$ ($p < 10^{-3}$) |
| Complaint Rate Index ($\downarrow$) | 1.000 | 0.865 | $[-15.7\%, -11.3\%]$ ($p < 10^{-3}$) |
| Subsidy Loss Index (n.s.) | 1.000 | 0.996 | $[-2.1\%, +1.3\%]$ ($p = 0.62$) |
| Daily Orders Served | | Millions | |

## 4.5 Online A/B Testing

To validate the real-world impact, GUARD was deployed on a large-scale e-commerce promotion platform and subjected to a 14-day online A/B test against the incumbent high-performing DNN model. **We calibrate the operating thresholds to match recall with the baseline** and report the resulting precision and complaint-related metrics (normalized due to data sensitivity) in Table 3. Under matched recall, GUARD improves scalper identification precision by **+8.9 points** (80.1% $\rightarrow$ 89.0%, 95% CI $[+7.9, +9.9]$, $p < 10^{-3}$ under a two-sided test on $n$=14 daily aggregates) and reduces the (normalized) complaint rate by **13.5%** (1.000 $\rightarrow$ 0.865, 95% CI $[-15.7\%, -11.3\%]$, $p < 10^{-3}$). We further monitor platform subsidy loss using daily aggregated statistics over the 14-day window and apply a standard two-sided significance test; the observed difference in normalized subsidy loss is $-0.4\%$ with 95% CI $[-2.1\%, +1.3\%]$ and $p = 0.62$, i.e., not statistically significant at the 5% level. The system now serves millions of daily orders, demonstrating its robustness and scalability in a live, high-stakes production environment. Beyond model metrics, the sustained reduction in complaint volume directly translates into recovered trust from legitimate users who had previously been blocked in error—the central operational goal that motivated this work.

## 5 Discussion and Generalization

**Why adversarial invariance outperforms naive data utilization.** A natural question is: why not simply add complaint-verified false positives as negative samples? Our experiments (Table 1, DNN + Conf. Neg.) show that this yields only modest gains (+2.3% precision). The reason is that adding negatives adjusts the *decision boundary* but does not change the *representation*: the encoder still maps confused benign users and scalpers to nearby regions in latent space, and the classifier must carve an increasingly complex boundary to separate them. In contrast, adversarial invariance reshapes the *geometry* of the representation space itself, pushing confused benign users away from scalper-like regions (as visualized in Figure 8). This explains why GUARD's gains are robust across operating thresholds (Figure 4).

**Precision–recall trade-off as a design choice.** GUARD improves precision at the cost of a slight recall reduction (82.1%$\rightarrow$80.2% in offline evaluation). This is the operationally correct trade-off for promotion risk control: false positives generate direct costs (complaint handling, compensation, user churn, reputation damage), whereas missed scalpers can be caught by downstream mechanisms such as settlement-stage audits and graph-based retrospective analysis. Crucially, when thresholds are calibrated to match recall in the online A/B test, GUARD still achieves +8.9 precision points, confirming that the gains stem from better-quality representations rather than a trivially shifted decision boundary.

**Hypothetical applications and required conditions (future work).** The core principle of GUARD—using verified post-decision feedback to define an adversarial domain for representation invariance—generalizes, *in principle*, to any classification system satisfying two conditions: (1) false positives are verifiable through user feedback or appeal mechanisms, and (2) false positives are *mechanism-driven* (concentrated on specific feature patterns) rather than random noise. We emphasize that the scenarios listed below are *illustrative future-work directions* where these two conditions plausibly hold; we do not claim empirical validation in these domains and leave such validation to subsequent work.

- **Content moderation (hypothetical):** wrongful content removals that are reversed on user appeal, where certain benign posting patterns (e.g., news reporting) overlap with policy-violating content.

- **Credit scoring (hypothetical):** loan application denials overturned on appeal, where legitimate applicants share features with high-risk borrowers (e.g., recent address changes due to job relocation).

- **Medical triage (hypothetical):** unnecessary referrals identified through follow-up, where healthy patients with certain symptom patterns are repeatedly over-triaged.

- **Ad fraud detection (hypothetical):** legitimate advertisers falsely flagged due to unusual but benign traffic patterns (e.g., viral campaigns).

In each case, the verified appeals/feedback would provide a natural Confusion Domain, and the adversarial invariance mechanism could in principle disentangle genuinely predictive factors from mechanism-driven confounders. Empirical confirmation in any of these domains is beyond the scope of this paper and is explicitly left as future work, consistent with the limitations stated in Sec. 6.

## 6 Broader Impact and Ethical Considerations

**Positive impact.** GUARD is explicitly designed to *protect* benign users from false accusations by automated risk-control systems. By reducing false positives in the confusion zone, it improves shopping experience for legitimate users who would otherwise face order cancellations, account restrictions, and degraded service. The 13.5% complaint rate reduction observed in the online A/B test represents a direct improvement in user welfare at scale.

**Platform power and fairness.** We acknowledge that automated risk-control systems create power asymmetry between platforms and users: platforms make unilateral blocking decisions that users must actively appeal to reverse. GUARD partially mitigates this asymmetry by leveraging the appeal mechanism as a feedback loop to *reduce* wrongful decisions. However, it does not eliminate the underlying structural asymmetry, and not all affected users file complaints. Future work should explore how to extend coverage to non-complaining confused users, potentially through proactive fairness audits.

**Promotion mechanisms and over-consumption.** Large-scale e-commerce promotions with deep subsidies can incentivize over-consumption and contribute to environmental costs. While GUARD does not directly address this concern (its scope is accurately distinguishing scalpers from benign users), we note that accurate detection helps ensure subsidies reach intended beneficiaries rather than arbitrage operators, which partially aligns platform incentives with consumer welfare.

**Limitations.** Our evaluation is conducted on a single platform's data, and while the method is designed to generalize (Sec. 5), empirical validation on other domains remains future work. Additionally, the Confusion Domain is constructed from users who actively complain, introducing a selection bias toward users who are aware of and willing to use appeal mechanisms. We note that the closed-loop reinforcement of false positives across retraining cycles, while a well-recognized concern in deployed risk-control systems Sculley et al. (2015), is not directly measured in this work. Our evaluation focuses on demonstrating that GUARD effectively reduces false positives within a single training–deployment cycle—the fundamental building block for breaking such feedback loops. A longitudinal study tracking FPR-B across multiple retraining iterations would provide additional evidence on whether GUARD's benefits compound over time by preventing false positives from re-entering the training pipeline as positive labels. We consider this a valuable direction for future work, particularly in combination with continual learning frameworks that explicitly model distribution shift across retraining cycles.

# 7    Conclusion

In this paper, we addressed the critical challenge of scalper detection in the presence of *instance-dependent* label noise caused by intrinsic feature overlap between scalpers and certain benign users in promotion scenarios. We shift from passive noise handling to *feedback-grounded* representation disentanglement by proposing GUARD, an adversarial learning framework that leverages *complaint-verified false positives* to define a Confusion Domain. By enforcing confusion-invariant representations through a GRL-based objective, GUARD learns risk-predictive features that are less sensitive to complaint-triggering superficial cues. To mitigate the scarcity of verified complaints, we further expand the Confusion Domain via uncertainty-based mining. Extensive offline experiments and a 14-day online A/B test on a large-scale e-commerce promotion platform demonstrate that GUARD substantially improves precision and reduces false positives; under thresholds calibrated to match recall, it further decreases the (normalized) complaint rate while keeping subsidy loss statistically unchanged. Overall, our work shows how post-decision user feedback can be systematically incorporated to improve robustness in high-stakes industrial risk control.

## Reproducibility Statement

We provide the following to support reproducibility:

- **Code availability:** The complete training code for GUARD is included as supplementary material (`guard_supplementary_code.zip`), organized as a self-contained Python package with the following structure: `models/guard_model.py` (shared-trunk dual-head architecture), `utils/grl.py` (Gradient Reversal Layer), `train.py` (full adversarial training loop with annealed implicit loss and MC Dropout mining), `ablation_study.py` (ablation experiments reproducing Table 2), `predict.py` (inference and evaluation), `experiments/public_dataset_benchmark.py` (runnable benchmark on public data demonstrating the full pipeline without proprietary data), `data/data_loader.py` (data loading with synthetic confusion-zone generation), `config/config.py` (all hyperparameters matching Section 4.2), and `run_all.sh` (one-command reproduction script). The code requires only standard PyTorch dependencies listed in `requirements.txt`. *Public release.* We commit to releasing this reference implementation and the synthetic-data generator under a permissive open-source license at the camera-ready stage, in a public repository linked from the camera-ready version of this paper, to facilitate independent verification and adoption.

- **Implementation details:** Section 4.2 provides comprehensive hyperparameters (learning rate $10^{-4}$, batch sizes 512/256/1024, trunk dimension 64, latent dimension 32, dropout rate 0.5, $\lambda = 0.4$, annealing schedule $\beta \in [0.01, 0.4]$ over 50 epochs, MC Dropout passes $T = 20$, confidence threshold $\tau = 0.9$, Top-$K = 16$, positive class weight 5, early stopping patience 8) sufficient for reproduction on similar tabular risk-control datasets.

- **Data constraints and order-of-magnitude disclosure:** The industrial dataset cannot be released due to proprietary business constraints and user privacy regulations (containing user behavioral features, device information, and transaction data subject to data protection laws). To partially mitigate this, Section 4.2's dataset description discloses order-of-magnitude statistics for $|\mathcal{D}_r|$, $|\mathcal{D}_c|$, $|\mathcal{U}|$, the test set, and the $B=1$ cohort, which do not violate confidentiality but allow assessment of statistical reliability. The method itself requires only three readily available components: (1) a labeled training set, (2) a set of verified false positives from any appeal or feedback mechanism, and (3) a pool of unlabeled flagged samples. These components exist in most deployed classification systems with user feedback channels. The supplementary code includes a synthetic data generator that produces data with mechanism-driven confusion zones matching the structural properties described in Section 1.

- **Applicability beyond our setting:** While we cannot release our specific industrial dataset, the core GUARD framework (GRL-based confusion invariance + MC Dropout mining) is architecture-agnostic and can be applied to any tabular or structured risk-control system. We describe the

minimal data requirements and integration steps in the supplementary README to facilitate adoption.

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
