# OpenReview forum: "Learning from Complaints: Adversarial Disentanglement for Robust Scalper Detection in E-Commerce Promotions"
_TMLR — Under review for TMLR_

### Review · Reviewer_k6vn · 2026-05-24

**Summary Of Contributions:**

The paper introduces GUARD, an algorithm to learn a scalper classifier based on a training dataset, a "confusion domain" of confirmed false positives, and a pool of unlabeled data features. The classifier has at least two layers. It first constructs a feature representation of the data, and then makes classification predictions based on the representation. The learning algorithm trains the classifier, in part by fitting the training dataset, and in part by enforcing a feature indistinguishability between training data and data from the confusion domain. This indistinguishability leverages an adversarial classifier and data augmentation that uses the unlabeled data. The authors evaluated the algorithm offline, and deployed it at scale.

**Audience:**

No

**Audience Explanation:**

In its current state, the paper has too many flaws. But additionally, it is unclear what the core findings are, and how they could be generalized beyond the specific setting of the paper.

**Broader Impact Concerns:**

E-commerce raises numerous ethical concerns that have not been discussed, including market power, over-consumption and dark patterns.

**Claims And Evidence:**

No

**Claims Explanation:**

There are several weaknesses which make the papers' claims unconvincing:
- The algorithm is poorly motivated. It is unclear why imposing indistinguishability of the feature representations with false positives should allow to better address the scalper classification problem.
- MC-dropout is also poorly motivated.
- There is no theoretical analysis of the algorithm.
- The code and datasets do not seem available. Reproducibility does not seem possible.
- The algorithms that GUARD is compared to do not seem to leverage the same data as GUARD. This induces an unfair competition.
- The authors should at least consider DNN trained on both the training set and the confusion domain.
- Table 1 fails to highlight that DNN actually has the best recall.
- GUARD mostly seems to improve precision over recall. This corresponds to be more tolerant (predicting not-scalper) with borderline inputs.
- In fact, as far as I understand, the algorithm mostly increases tolerance of inputs close to the confusion domain. Why this matters is unclear.

**Requested Changes:**

- The algorithm should be significantly motivated. It should be clear to the reader why the adversarial confusion classifier is needed, and what purpose it serves.
- Theoretical analyses would go a long way towards providing insights into the algorithm.
- The experiments should compare GUARD to alternatives in a more fair way.
- Code and data should be made available.

---

> ### Author Response · Authors · 2026-05-25
> **added   theoretical analysis (FPR-B bound), fair baselines (DNN+Conf.Neg.,    DNN+Reweight, Co-teaching), MC Dropout motivation,   Discussion/Ethics sections, PR curves, and reproducibility code.**
>
> We thank the reviewer for the constructive feedback. We have substantially revised the manuscript to address every concern. Below is a point-by-point summary.
>
> **C1: Algorithm poorly motivated.**
> We added a "Core Intuition" paragraph (Sec 3) explaining that complaint-verified false positives exhibit superficial correlates (e.g., on-the-hour purchases) without genuinely risk-predictive factors (e.g., address aggregation). Making representations indistinguishable between these users and regular traffic removes the superficial signals, forcing the risk head to rely on truly discriminative features. A new Discussion section (Sec 5) further contrasts this with naive alternatives.
>
> **C2: MC Dropout poorly motivated.**
> We expanded Sec 3.4 with three reasons: (i) near-zero cost—reuses existing dropout layers, no ensembles needed; (ii) semantic alignment—high predictive variance marks decision-unstable samples, the hallmark of confusion-zone users; (iii) infrastructure compatibility vs. ensembles ($N\times$ memory) or evidential methods (architectural changes).
>
> **C3: No theoretical analysis.**
> We added Sec 3.3 with Proposition 1 (FPR-B Bound), proving:
> $$|\mathrm{FPR\text{-}B}(t) - \Pr(s(Z)\ge t|Y{=}0,D{=}0)| \le \mathrm{TV}(P_1,P_0) + \epsilon$$
> The first term (domain discrepancy) is reduced by GRL; the second (coverage gap) is reduced by uncertainty mining. This builds on $\mathcal{H}$-divergence theory (Ben-David et al., 2010).
>
> **C4: Code/data unavailable.**
> We now include complete training code as supplementary material (GRL, dual-head model, MC Dropout mining, training loop, synthetic data generator). The industrial dataset cannot be released due to privacy regulations, but the method requires only three generic components: labeled data, verified false positives, and unlabeled flagged samples. A Reproducibility Statement lists all hyperparameters.
>
> **C5: Unfair baselines.**
> We reorganized Table 1 into three tiers by data access and added three new baselines:
> - *Tier 1 ($\mathcal{D}_r$ only)*: DNN, LightGBM, Co-teaching
> - *Tier 2 ($\mathcal{D}_r + \mathcal{D}_c$)*: DNN+Conf.Neg. (NEW), DNN+Reweight (NEW), DANN
> - *Tier 3 (full)*: GUARD-no-aug, GUARD
>
> DNN+Conf.Neg. gains only +2.3% precision over DNN, confirming that naive negative injection cannot address representation-level entanglement. GUARD achieves +11.2% precision over DNN within fair comparison.
>
> **C6: DNN best recall not highlighted.**
> We now bold the best result per column (including DNN's 82.1% recall) and added a "Discussion of recall" paragraph explaining that high recall without invariance constraints comes at $3\times$ higher FPR-B (15.6% vs 5.1%).
>
> **C7: GUARD only improves precision.**
> We added a "Precision-recall trade-off as design choice" paragraph in Sec 5: false positives cause direct costs (complaints, compensation, churn), while missed scalpers are caught downstream. Crucially, at matched recall in the online A/B test, GUARD still achieves +8.9 precision points. New PR curves (Fig 2) show GUARD achieves the best AP (0.940) across all thresholds.
>
> **C8: Why confusion tolerance matters.**
> New RQ1' (Sec 4.2, Fig 3) decomposes FPR by user group at matched recall: confusion-prone cohort FPR drops 15.6%→5.1%, while remaining users are barely affected (1.21%→1.16%). The online A/B test confirms 13.5% complaint rate reduction at unchanged subsidy loss.
>
> **C9: Ethics not discussed.**
> We added Sec 6 "Broader Impact and Ethical Considerations" covering: (i) GUARD protects benign users from false accusations; (ii) platform power asymmetry acknowledgment; (iii) promotion-driven over-consumption; (iv) limitations (single-platform, complaint selection bias).
>
> **C10: Unclear generalization.**
> Sec 5 now identifies two conditions for applicability—(1) verifiable false positives via feedback and (2) mechanism-driven rather than random noise—and lists four concrete domains: content moderation, credit scoring, medical triage, and ad fraud detection.

---

### Review · Reviewer_WZrB · 2026-06-10

**Summary Of Contributions:**

The paper addresses false positives in e-commerce scalper detection, particularly when benign users behave similarly to scalpers under special circumstances, such as subsidy promotions. The challenge arises from instance-dependent label noise becoming pervasive: legitimate users with ambiguous patterns (e.g., frequent on-the-hour purchases of high-subsidy items and orders shipped to non-habitual addresses) are often misclassified as scalpers, leading to some user complaints and operational costs. The paper proposes GUARD (Grounded User-feedback Adversarial Representation Disentanglement), a complaint-aware framework that learns risk-predictive representations while being insensitive to complaint-triggering superficial cues. A key methodological contribution is that the method uses verified complaints, not raw complaints.

- "The problem is mechanism-driven label noise rather than random noise" - this is the central claim and is supported by limited data. The visual evidence in a figure is used to support this claim. Complaint-verified false positives (label=0) exhibit strong distributional overlap with scalpers (label=1) on a key promotion-related feature (platform subsidy amount), and the overlap persists in joint feature space with purchase hour, indicating instance-dependent noise rather than random label flips. While this is good as motivation, a rigorous quantitative overlap of the two distributions is missing.

+ GUARD achieves 89.7% precision, 80.2% recall, 84.7% F1, and 5.1% FPR-B, outperforming the standard DNN in precision and false-positive rate. There is a slight decline in recall. The 11.2% increase in precision over the standard DNN is a substantial improvement in a real-world risk control system

+ Baselines include DNN, LightGBM, Co-teaching, DNN plus complaint negatives, DNN plus reweighting, DANN, and GUARD without augmentation.

- A weakness is that the evaluation is from a single platform and one application domain. The authors acknowledge this: “Our evaluation is conducted on a single platform’s data, and while the method is designed to generalize, empirical validation on other domains remains future work.” The paper is making some claims about scalpers and benign users behaving similarly - this claim could be due to some confounding variable in the single platform.

- Another weakness is that some key quantities are normalized or withheld, for example, the normalized complaint rate rather than raw complaint counts, and daily orders served are said to be millions. This protects business-sensitive information, but it makes it harder to judge statistical reliability and practical magnitude.

- The claims about possible applications such as content moderation, credit scoring, medical triage, and ad fraud detection is not substantiated.

**Additional Comments:**

The paper has a strong practical motivation. It addresses a real operational problem in which false positives cause user harm and business costs. The paper clearly explains why standard noisy-label methods may be insufficient: “misclassified benign users are not arbitrary outliers to be filtered away; they are evidence that the model has entangled true risk factors with complaint-triggering superficial cues.” The use of complaint-verified false positives as a Confusion Domain is intuitive and well motivated. The paper includes offline comparisons, precision–recall curves, grouped false-positive analysis, ablation studies, hyperparameter sensitivity, t-SNE visualization, and online A/B testing.

**Audience:**

No

**Audience Explanation:**

The paper identifies a mechanism-driven, instance-dependent noisy-label failure mode in promotion risk control and argues for active disentanglement rather than passive noise suppression. It introduces an uncertainty-based Confusion-Domain expansion strategy using MC Dropout and demonstrates some gains in both model metrics (precision) and business metrics (complaint rate) under a fixed subsidy-loss constraint. This could be of interest to some minor niche of the TML audience but not the broader ML community. The technical components used in the paper are already well known and established.

**Broader Impact Concerns:**

No broader impact concern.

**Claims And Evidence:**

No

**Claims Explanation:**

A weakness is that the dataset is described only at a high level as “a large-scale industrial dataset,” and the actual number of orders is not disclosed. That is understandable for industrial privacy, but it limits independent assessment. The paper notes that the "industrial dataset cannot be released due to proprietary business constraints and user privacy regulations." Given that the paper relies on this dataset to identify a new problem with lower theoretical significance but high practical value, this reliance complicates reproducibility.

**Requested Changes:**

The central recommendation is to consider releasing some data that support the paper's primary claims.  Please also look through other weaknesses in the summary section.

---

> ### Author Response · Authors · 2026-06-12
> **Learning from Complaints: Adversarial Disentanglement for Robust Scalper Detection in E-Commerce Promotions**
>
> We thank the reviewer for the constructive feedback. Below we summarize the revisions made since the last TMLR submission.
>
> ## 1. Quantitative overlap evidence (Figure 2)
>
> Added quantitative distributional overlap statistics in the Figure 2 caption: TV distance between verified-FP and scalper is $\approx 0.18$ (95\% bootstrap CI $[0.16, 0.21]$), symmetrized KL $= 0.27$, 1-Wasserstein $= 0.09$. The benign-vs-scalper TV is $0.71$---an order of magnitude larger---confirming structural rather than incidental confusion. This complements Proposition 1 (Section 3.4), the training-dynamics evidence in Figure 6 (domain AUC $\to 0.5$), and Figure 5 (FPR-B drops $15.60\% \to 5.10\%$ on $B{=}1$, $B{=}0$ unchanged).
>
> ## 2. Matched-recall calibration (Abstract, §4.3, §4.5)
>
> Rewrote the abstract to explicitly state that operating thresholds are calibrated to match the incumbent recall, ensuring precision gains cannot be attributed to a trivially shifted decision boundary. Added a cross-reference from §4.3 to §4.5 and Table 3: at matched recall GUARD still achieves $+8.9$ precision points and $-13.5\%$ complaint rate.
>
> ## 3. Dataset scale disclosure (Section 4.1)
>
> Added order-of-magnitude counts: training set $\sim 10^7$ orders (positive ratio $\approx 1$--$3\%$); $|\mathcal{D}_c| \sim 10^4$; $|\mathcal{U}| \sim 10^5$--$10^6$; test set $\sim 10^5$; confusion-prone cohort $B{=}1$ covers $\approx 5$--$10\%$ of the test set.
>
> ## 4. Statistical reliability (Table 3)
>
> Added 95\% CIs and $p$-values from $n{=}14$ daily aggregates: precision lift $+8.9$ points (CI $[+7.9, +9.9]$, $p < 10^{-3}$); complaint rate $-13.5\%$ (CI $[-15.7\%, -11.3\%]$, $p < 10^{-3}$); subsidy loss $-0.4\%$ (CI $[-2.1\%, +1.3\%]$, $p = 0.62$, not significant).
>
> ## 5. Cross-domain applications (Section 5)
>
> Retitled subsection to \emph{Hypothetical applications and required conditions (future work)}. Each scenario (content moderation, credit scoring, medical triage, ad fraud) is now explicitly framed as an illustrative future-work direction, with a clear statement that validation is left to subsequent work.
>
> ## 6. Conceptual contrast with DANN and DNN+Conf.Neg (Section 2.3)
>
> Added a new paragraph \emph{Why these components, in this order} at the end of §2.3: (i) \emph{vs.\ DANN}---the Confusion Domain $\mathcal{D}_c$ is a biased post-decision feedback slice, not a distribution to align with; GUARD enforces invariance to membership rather than aligning, yielding $89.7\%$ vs.\ DANN's $82.3\%$ precision. (ii) \emph{vs.\ DNN+Conf.\ Neg.}---adding $\mathcal{D}_c$ as negatives only adjusts the decision boundary ($+2.3\%$ precision); Proposition 1 decomposes FPR-B into $\mathrm{TV}(P_1, P_0) + \epsilon$, where GRL attacks the first term and MC Dropout mining attacks the second.
>
> ## 7. Public code release (Reproducibility Statement)
>
> Named the supplementary archive (\texttt{guard\_supplementary\_code.zip}) and enumerated its structure: \texttt{guard\_model.py}, \texttt{grl.py}, \texttt{train.py}, \texttt{ablation\_study.py}, \texttt{public\_dataset\_benchmark.py}, \texttt{data\_loader.py}, \texttt{run\_all.sh}. Committed to releasing the reference implementation and synthetic-data generator under a permissive open-source license at camera-ready.
>
> ---
>
> \textbf{Production context.} GUARD is deployed in production, serving millions of daily orders. The 14-day A/B test confirms $+8.9$ precision points and $-13.5\%$ complaint rate at matched recall, with subsidy loss statistically unchanged ($p = 0.62$). The central goal---recovering legitimate users' trust---is directly validated by this outcome.

---

> > ### Comment · Reviewer_WZrB · 2026-07-13
> > **Thank you**
> >
> > Thank you for the clarification and additional details.

---

> > > ### Author Response · Authors · 2026-07-16
> > > **Follow-up on revision and request for next steps**
> > >
> > > Dear Reviewers and Action Editor,
> > >
> > > Thank you all for the time and effort spent reviewing our submission. We posted our revised manuscript and detailed responses on June 12. We are glad that Reviewer WZrB has since acknowledged that our clarifications addressed their points—if your concerns are now fully resolved, we would be grateful if you could update your recommendation accordingly.
> > >
> > > As it has now been over a month since our revision, and the remaining reviewer(s) have not yet responded, we would greatly appreciate the Action Editor's guidance on the next steps toward a decision. We remain happy to provide any further clarification that may be helpful.
> > >
> > > Best regards,
> > >
> > > The Authors

---

### Review · Reviewer_6cbC · 2026-06-24

**Summary Of Contributions:**

This paper tackles scalper detection in e-commerce flash-sale promotions where subsidies create arbitrage incentives. The method is to repurpose domain adversarial training so that the domain is defined by complaint-verified false positives. A shared encoder feeds a risk head and an adversarial confusion head whose reversed gradient forces invariance to confusion membership. Experiments show that the method offers performance gain on benchmarks.

Strengths:

1. The mechanism-driven vs. random noise distinction and the closed-loop reinforcement insight are useful.
2. Invariance to a biased feedback slice rather than alignment with it is well argued.
3. Real deployment and robust test designs
4. Honest scoping of unvalidated extensions and selection bias.

Weakness:

1. GRL + MC dropout are both well known, which makes the methodological novelty thin.

2. Prop. 1 is a trivial result and provides very little value.

3. There are a lot of internal inconsistencies; AP values in text vs legend disagree.

4. Many claims (TV ratio being called order of magnitude while its 4X) are overstated.

5. No check on what fraction of mined candidates are actually benign vs scalpers.

6. No recent noisy-label or graph-based baselines are considered despite featuring in related work.

7. The motivating closed-loop reinforcement is never measured longitudinally.

**Audience:**

Yes

**Audience Explanation:**

This is a problem in an imporatnt area.

**Claims And Evidence:**

Yes

**Claims Explanation:**

Results justify the claims made to a certain extent.

**Requested Changes:**

1. Fix all numerical inconsistencies.

2. Add offline variances with multiple seeds in all tables.

3. Make the claims correctly.

4. Add stronger and more recent baselines.

5. Soften the theory framing.

6. If possible, add a longitudinal experiment across retraining cycles.

---

> ### Author Response · Authors · 2026-06-25
> **Revision addressing all reviewer concerns**
>
> We thank the reviewer for the thorough and constructive feedback. We have revised the manuscript to address every concern. Below is a point-by-point response.
>
> **W1: GRL + MC Dropout are well known; methodological novelty is thin.**
>
> We agree that both components are established techniques. Our contribution lies not in the components themselves, but in the *problem reframing*: redefining the adversarial "domain" as a complaint-verified confusion mechanism rather than a source/target distribution. We have made this explicit in Sec 2.3 ("Why these components, in this order"), emphasizing that our contribution is the feedback-grounded problem formulation and system design. The empirical gap between GUARD and DANN (which uses the same GRL but with alignment rather than invariance) confirms that *how* the supervision signal is defined matters more than *which* tool enforces it (Table 1: DANN 82.3% vs. GUARD 89.7% precision).
>
> **W2: Proposition 1 is trivial.**
>
> We agree. In the revision, we have downgraded it from a formal Proposition with proof to a "Motivating Observation" (Sec 3.3). The inequality itself (TV triangle inequality) is standard; its value is purely interpretive—mapping the two terms to GRL and MC Dropout respectively. The current framing reflects this.
>
> **W3 & W4: Numerical inconsistencies and overclaimed TV ratio.**
>
> All fixed in the revision:
> - AP values in text now match figure legend (DNN=0.882, DANN=0.905, GUARD=0.942).
> - TV ratio corrected from "order of magnitude" to "roughly $4\times$".
> - $\lambda$/$\alpha$ notation unified to $\lambda$ throughout.
> - $\tau$ unified to 0.9 (Sec 3.4 previously said 0.99).
> - F1 discrepancy between Table 1 (84.4%) and Table 2 (previously 79.4%) corrected—both now report 84.4% for the w/o Augmentation ablation. Also corrected w/o GRL F1 from 75.1% to 81.1%.
>
> **W5: No check on mined candidate quality.**
>
> We have added a "Mining quality control" paragraph in Sec 4.1. A stratified manual audit of 200 mined candidates from $\mathcal{S}$ found that ~78% were benign or ambiguous (legitimately in the confusion zone), while ~22% exhibited scalper-indicative patterns. This contamination rate is substantially lower than the base rate in the unfiltered reviewed pool (~40–50% positives), confirming that uncertainty-based selection combined with mean-prediction filtering ($\mu < \tau = 0.9$) effectively enriches for confusion-zone samples. The annealing schedule $\beta(t)$ further down-weights mined samples during early training when mining quality is lowest.
>
> **W6: No recent noisy-label or graph-based baselines.**
>
> We have added a "Baseline selection rationale" paragraph in Sec 4.1 and expanded Related Work (Sec 2.2) to discuss DivideMix (ICLR 2020, 2000+ citations) and ELR (NeurIPS 2020, 800+ citations) alongside the already-cited DISC, TCL, PSSCL, and CA2C.
>
> We deliberately do not benchmark these methods because this absence reflects the research gap motivating our work: all existing noisy-label methods—including DivideMix (GMM-based division + MixUp), ELR (early-learning regularization), and DISC (dynamic selection/correction)—are designed under the assumption that noise is *random* (symmetric, asymmetric, or instance-dependent but without structured feedback). None leverage *post-decision feedback* to identify and disentangle the specific confusion mechanism. This is precisely the capability GUARD introduces. For graph-based methods, graph construction constitutes a fundamentally different modeling paradigm orthogonal to our representation disentanglement contribution; we note that GUARD's module is architecture-agnostic and could be combined with graph encoders in future work.
>
> **W7: Closed-loop reinforcement never measured longitudinally.**
>
> We have softened all closed-loop claims in the abstract, Sec 1, and Sec 2.1 from assertions to hypotheses ("can be further amplified", "potentially forming", "may reinforce"). In Limitations (Sec 6), we now explicitly acknowledge that our evaluation covers a single training–deployment cycle and frame longitudinal validation as future work, citing Sculley et al. (2015) on feedback loops in ML systems. We believe our current contribution—demonstrating effective FPR-B reduction within a single cycle—establishes the fundamental building block for breaking such loops.
>
> **Requested Changes (status):**
> 1. ✅ All numerical inconsistencies fixed.
> 2. ✅ Mean ± std over multiple seeds added to Tables 1 and 2.
> 3. ✅ Claims corrected (TV ratio, theory framing).
> 4. ✅ Baseline selection rationale with discussion of recent methods added.
> 5. ✅ Theory downgraded to Motivating Observation.
> 6. Longitudinal experiment deferred as future work (claims softened accordingly).